

# Impacts of Environmental Conditions and Ice Nuclei Recycling on Arctic Mixed-Phase Cloud Properties

Benjamin Ascher[1] and Fabian Hoffmann[2]

[1]Meteorological Institute Munich, Ludwig Maximilian University, Munich, Germany
[2]Institute for Meteorology, Free University of Berlin, Berlin, Germany

**Correspondence:** Benjamin Ascher (ben.ascher@lmu.de) and Fabian Hoffmann (f.hoffmann@fu-berlin.de)

**Abstract.** Radiatively driven Arctic stratocumulus clouds have important climactic impacts due to their effects on surface radiative balance. The presence of both liquid and ice within Arctic stratocumulus, and their interaction through the Wegener-Bergeron-Findeisen (WBF) process, strongly affects the properties and lifetimes of these clouds. To assess the impacts of mixed-phase microphysical processes in Arctic Stratocumulus, we use a Langrangian cloud microphysical model within a large eddy simulation framework to simulate a single-layer cloud under varying free-tropospheric humidity and above-cloud inversion strength. We also run two simulations in which precipitating ice crystals have their ice nuclei (IN) re-injected into the model domain, rather than removed. We find that IN recycling plays a critical role in maintaining the presence of ice in the mixed-phase cloud. The simulations with drier free-tropospheric air experience greater sublimation of ice crystals below cloud, recycling of ice crystals, and a higher ice water path than simulations with more humid free-tropospheric air. We also find that the impact of inversion strength on cloud microphysical characteristics is strongly modulated by free-tropospheric relative humidity, with decreased inversion strength resulting in both increased and decreased liquid water path under high and low free-tropospheric relative humidity, respectively.

## 1 Introduction

Stratocumulus clouds, weakly convective shallow cloud systems, occur frequently over large areas of the Earth's surface across many different latitudinal and climate regimes (Shi et al., 2018; Wood, 2012). These long-lived cloud fields play a critical role in the climate system, as their high albedo reflects a large portion of incoming solar radiation, resulting in a cooling effect during the day. Their impact on Earth's radiative balance, and investigation into how this impact may change due to anthropogenic climate and aerosol forcing, has been extensively studied (Caldwell and Bretherton, 2009; Tan and Storelvmo, 2019).

Compared to stratocumulus clouds in the subtropics, Arctic stratocumulus clouds have important impacts on the climate system not just as a result of their reflection of solar radiation back into space, but also from their emission of longwave radiation downward to the surface (Eirund et al., 2020; Garrett et al., 2002; Tan and Storelvmo, 2019; Tan et al., 2023). With the rapid rate of Arctic warming as a result of climate change, a better understanding of Arctic stratocumulus, and their influence on Arctic energy balance, is thus a research area of significant importance.





While subtropical stratocumulus clouds are entirely composed of liquid, Arctic stratocumulus clouds are complicated by
25 the presence of ice (Mioche et al., 2017; McFarquhar et al., 2011; Shupe et al., 2006; Verlinde et al., 2007). Due to the lower
equilibrium vapor pressure over ice compared to liquid at the same temperature, ice crystals within Arctic stratocumulus clouds
grow much more rapidly than liquid cloud droplets, effectively competing for water vapor and reducing the liquid water path
(LWP) of these clouds, an effect known as the Wegener-Bergeron-Findeisen (WBF) process (Harrington et al., 1999).

Since Arctic stratocumulus clouds are maintained by radiative cooling at the cloud top, this reduction in LWP and concomi-
30 tant reduction in cloud-top radiative cooling can weaken the boundary-layer circulations which maintain the cloud. If enough
ice is present, this can lead to a destructive feedback loop and the eventual dissipation of the cloud entirely. Even if the cloud
persists, the reduction of LWP which is a consequence of the presence of ice alters the downward flux of longwave radiation
to the surface beneath the cloud. The ice crystal number concentration, which strongly affects LWP and ice water path (IWP)
within the clouds, is heavily dependent on the number of ice nuclei (IN) within the cloud. Larger IN concentrations have been
shown both in modeling and observational studies to increase IWP at the expense of LWP, sometimes leading to complete
glaciation and dissipation of the cloud (Ahola et al., 2020; Das et al., 2025; Ovchinnikov et al., 2014; Solomon et al., 2018).

Arctic stratocumulus clouds also often precipitate ice, which stabilizes the boundary layer by cooling as falling ice crystals
sublimate below the cloud. This boundary layer stabilization, as well as the production of cold pools, plays an important role in
determining the morphology and organization of Arctic stratocumulus clouds (Eirund et al., 2019; Sotiropoulou et al., 2014).
Due to the importance of ice processes within Arctic stratocumulus to the morphology and lifetimes of these clouds, as
well as their radiative impacts on the surface, these clouds have been the subject of both modeling and observational studies
(Ahola et al., 2020; Avramov and Harrington, 2010; de Boer et al., 2009; Eirund et al., 2019; Fan et al., 2009; Harrington et al.,
1999; McFarquhar et al., 2007, 2011; Ovchinnikov et al., 2014; Paukert and Hoose, 2014; Raatikainen et al., 2022; Shupe et
al., 2006, 2011; Solomon et al., 2015, 2018; Verlinde et al., 2007). However, despite the insights gained from these projects,
numerous unknowns remain regarding ice growth within Arctic stratocumulus and its effects on the cloud as a whole. While
many processes affect the growth rate and number concentration of ice crystals in Arctic stratocumulus, including aggregation,
riming, secondary ice production from ice-ice collisions and rime splintering, we will focus primarily on ice growth through
vapor deposition (Jackson, 2012; Sotiropoulou et al., 2020). In particular, we will examine the roles of environmental humidity,
entrainment strength, and IN recycling on vapor depositional growth of ice.
The role of entrainment and mixing on ice crystal growth within Arctic stratocumulus clouds is another outstanding source of
uncertainty in the prediction of microphysical properties of these clouds. Arctic stratocumulus clouds typically occur in shallow
boundary layers beneath a drier and warmer free atmosphere above (McFarquhar et al., 2007, 2011; Mioche et al., 2017; Shupe
et al., 2006; Verlinde et al., 2007). At the top of the cloud, entrainment of this warm, dry air into the cloud can occur. As a result
of the WBF process, dry air entrainment in a mixed-phase cloud impacts liquid and ice particles quite differently. Hoffmann
(2020) suggests that such dry air entrainment could enhance the growth of ice crystals, as rapid evaporation of cloud droplets
exposed to subsaturated air provides a ready source of water vapor which can then be consumed by ice crystals. Observations
of single-layer Arctic stratocumulus, however, consistently find a predominantly-liquid layer near cloud top, suggesting that
WBF processes may be less important at cloud top than sedimentation of ice crystals and the production of supersaturation





through longwave radiative cooling (de Boer et al., 2009; Griesche et al., 2021; McFarquhar et al., 2007, 2011; Mioche et al.,
2017; Shupe et al., 2006; Verlinde et al., 2007). The rate of entrainment of air from the entrainment layer into the cloud is highly
dependent on the magnitude of the temperature difference across the entrainment layer, with smaller temperature differences
(weaker inversion strengths) being conducive to greater entrainment rates (Randall, 1984; Wood, 2012; Xu and Xue, 2015). The
effects of entrainment on cloud microphysical properties are also strongly influenced by the relative humidity above the cloud,
meaning that entrainment can alternately act to contribute to or deplete from the total water budget of Arctic stratocumulus
clouds (Egerer et al., 2021; Solomon et al., 2014). How these different inversion strengths and relative humidities interact,
particularly in the presence of ice, has not been well studied despite its potentially significant impacts on cloud properties and
radiative balance.

The goal of this study is therefore to examine the following two science questions relating to microphysical processes within
Arctic stratocumulus clouds. 1) How does the relative humidity above the cloud layer affect the microphysical properties within
Arctic stratocumulus clouds? 2) How does the strength of the above-cloud temperature inversion affect the entrainment rate
and microphysical processes within Arctic Stratocumulus clouds?

## 2 Methods

To answer the above questions, we chose to simulate a single-layer mixed-phase Arctic stratocumulus cloud field with the
System for Atmospheric Modeling (Khairoutdinov and Randall, 2003), an anelastic, nonhydrostatic Large Eddy Simulation
(LES) model. This model was run with a Lagrangian cloud microphysics model (Hoffmann et al., 2015), similar to the Super-
Droplet Method (SDM) of Shima et al. (2009, 2020). In the SDM approach, aerosol particles and hydrometeors are represented
as Super-Droplets (SDs), collections of many identical particles. A single gridbox may contain anywhere from dozens to
hundreds of these SDs. Each SD has a weighting factor, indicating how many physical aerosol particles/hydrometeors are
represented by the SD; extensive quantities, including total water mass, total aerosol mass, total ice nucleating particle (INP)
surface area; and intensive quantities such as dimensions (if an ice crystal), freezing temperature, and time spent within ice or
liquid supersaturated conditions. Each SD also has a unique ID assigned either at the start of the simulation or upon emission
(if a particle emission source is present in the domain).

The growth of ice crystals follows the approach of Chen and Lamb (1994). This method treats ice crystals as spheroids with
oblate and prolate axes (denoted as a- and c-axes, respectively). This parameterization uses an Inherent Growth Function (IGF)
to determine which shape ice crystals growing by vapor deposition should develop. This IGF, based primarily on laboratory
studies of ice crystal growth in free-fall chambers by Takahashi et al. (1991) are purely a function of temperature. This has the
consequence that, at a given temperature, two ice particles growing by vapor deposition under ice supersaturated conditions will
develop the same habits (though at different rates, depending on the magnitude of ice supersaturation). This parameterization
also calculates an effective deposition density and ventilation coefficient for ice crystals in an attempt to capture the impacts of
faceting and hollowing during vapor depositional growth and fall speed through vapor, respectively.



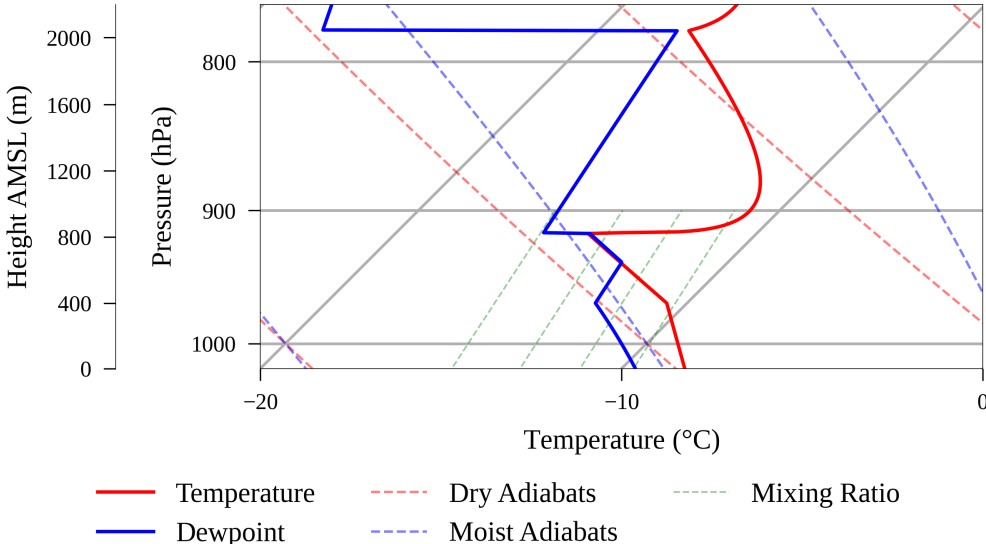

**Figure 1.** Initial thermodynamic conditions of the CONTROL simulation. Temperature and dewpoint are indicated by the red and blue solid lines, respectively, while dry and moist adiabats are indicated by the red and blue dashed lines, respectively. Isopleths of constant water vapor mixing ratio are depicted from $1.0 \, \mathrm{g \, kg^{-1}}$ to $1.6 \, \mathrm{g \, kg^{-1}}$, at intervals of $0.2 \, \mathrm{g \, kg^{-1}}$, in the green dashed lines.

As we are interested in examining the behavior of ice crystals under vapor deposition with little impact from collisional processes, we chose to simulate a single-layer precipitating mixed-phase Arctic stratocumulus cloud observed on 26 April 2008 during Flight 31 of the Indirect and Semi-Direct Aerosol Campaign (ISDAC, McFarquhar et al. (2011)). This cloud was observed both by ground-level remote sensing instruments and in situ aircraft observations. Observations of this day indicated

that the majority of ice crystals were pristine, with little indications of riming or aggregation (Jackson, 2012). Additionally, the single-layer nature of the cloud precludes seeder-feeder effects and reduces the chances of ice-ice collisions.

Initial environmental conditions are identical to those used in Ovchinnikov et al. (2014) and depicted here in Skew-T format in Figure 1. The boundary layer is initially decoupled from the surface, as evidenced by the stable lapse rate between the surface and 400 m. Between 400–650 m, there is well-mixed (dry adiabatic) layer, while between 650 and 820 m, the atmosphere

follows a moist adiabatic lapse rate. We also initialize all simulations with condensate following an adiabatic LWP profile between 650–820 m (not shown). The maximum initial value of liquid water mixing ratio is $0.163 \, \mathrm{g \, kg^{-1}}$ at 820 m. Zonal wind is initially set to $-7 \, \mathrm{m \, s^{-1}}$ throughout the domain, while meridional wind varies linearly with height from $-2 \, \mathrm{m \, s^{-1}}$ at the surface to $4.735 \, \mathrm{m \, s^{-1}}$ at 2250 m (here, negative values for zonal and meridional wind indicate easterly and northerly winds, respectively). Initial temperatures are approximately $-8.7 \, °\mathrm{C}$ at the surface, between $-13 \, °\mathrm{C}$ and $-15 \, °\mathrm{C}$ within the

cloud, and $-10.5°\mathrm{C}$ at the maximum of the above-cloud inversion.



All simulations use 10 meter vertical and 20 meter horizontal grid spacing, with 144 vertical levels and 64 grid points in each horizontal dimension. Horizontal boundaries are periodic, with a rigid model top at 1.44 km. Simulations are run with a 0.5 second timestep for 8 hours. Subgridscale turbulence is parameterized with a 1.5 order TKE-based closure, while radiation is parameterized using a Liquid Water Content (LWC)-profile approach as in Ovchinnikov et al. (2014). Surface energy fluxes

are neglected as in Ovchinnikov et al. (2014). Initial CCN number mixing ratio is 200 mg$^{-1}$ ($\sim$267 cm$^{-3}$ at the lowest model level) and initial IN number concentration is 1 L$^{-1}$. To reduce computation time, we only add SDs up to 1200 m of altitude.

We do not permit ice formation at the start of the simulation, but rather restrict it to occur beginning one hour after the start of the simulation. This is to permit a radiatively driven cloud circulation and STBL to begin to form without competition from ice crystals for water vapor. After one hour, SDs containing IN which are in liquid-supersaturated conditions undergo probabilistic

immersion freezing. The probability of a given SD freezing is calculated using a parameterization based on the behavior of SNOWMAX bacteria (Wex et al., 2015). SNOWMAX bacteria become active INP at approximately $-5$ °C and maintain approximately the same ice nucleating activity between $-10$ and $-20$ °C. Because temperatures within the cloud during this case are between $-13$ and $-15$ °C, the use of a SNOWMAX parameterization for ice nucleating activity therefore ensures that most IN-containing particles freeze within only a couple of timesteps after they enter liquid-supersaturated conditions. Once a

particle has nucleated, its SD type is instantly set to "ice" rather than "liquid". The ice nucleation parameterization used in this study takes no account of the amount of IN contained within an SD, but rather uses a simple "IN/No IN" binary classification for SDs. Future modeling work will explore the sensitivity of cloud microphysical evolution to different species or sizes of IN as described in Fu and Xue (2017).

To test the sensitivity of particle growth and cloud properties to both environmental conditions and microphysical parame-

125 terization, we present results from six simulations. Initially, these were a CONTROL case using the same initial atmospheric profile as that used in Ovchinnikov et al. (2014), a DRY case in which the water vapor mixing ratio above 825 m was reduced from 1.2 to 0.8 g kg$^{-1}$, a WEAKINV case in which the inversion strength was reduced from 3.5 K to 1.5 K, and a DRY_WEAKINV case combining the weaker inversion strength of WEAKINV and the reduced above-cloud mixing ratio of DRY. During the course of our analysis, we also discovered that depletion of IN through particle sedimentation was an impor-

130 tant factor in controlling both ice and liquid growth rates in all simulations. Therefore, we ran an additional two simulations, one with CONTROL environmental conditions and one with DRY conditions, in which any particle which reached the surface was re-injected into the below-cloud boundary layer as a dry CCN/IN particle. These simulations are denoted as INFLUX. Table 1 illustrates the full set of simulations.

## 3 Results

### 3.1 CONTROL

In the CONTROL simulation, a cloud is present from the start of the model run, but in an unrealistic state, with very little vertical motion. This initial state takes some time to spin up, as observed by the reduction in cloud liquid water mixing ratio



**Table 1.** Simulation parameters. "Above-Cloud Water Vapor Mixing Ratio" refers to the above-cloud water mixing ratio at initialization. "Inversion Strength" refers to the magnitude of the initial above-cloud temperature inversion. "IN Re-Injection" indicates whether IN-containing superparticles are removed from the simulation (None) or placed back in the simulation domain (Active) when an IN-containing superparticle reaches the surface.

| Simulation Name | Above-Cloud Water Vapor Mixing Ratio (g kg$^{-1}$) | Inversion Strength (K) | IN Re-Injection |
|---|---|---|---|
| CONTROL | 1.2 | 3.5 | None |
| DRY | 0.8 | 3.5 | None |
| WEAKINV | 1.2 | 1.5 | None |
| DRY_WEAKINV | 0.8 | 1.5 | None |
| INFLUX | 1.2 | 3.5 | Active |
| DRY_INFLUX | 0.8 | 3.5 | Active |

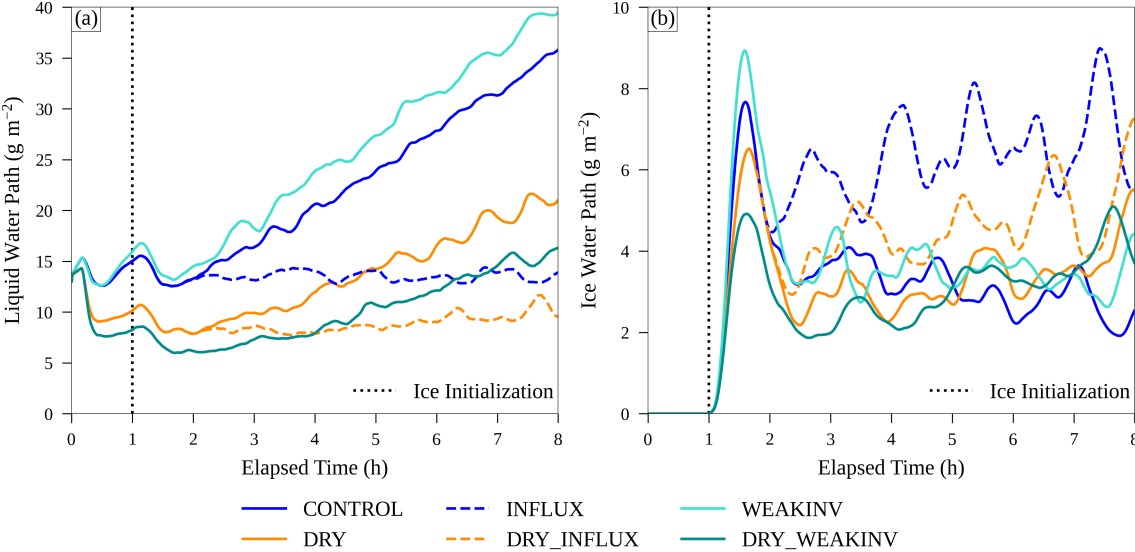

**Figure 2.** Time series of LWP (a) and IWP (b) in six simulations. The initiation of ice formation at $t = 1$ hour is indicated by the vertical dotted line.

at approximately time $t = 10$ minutes into the simulation. After $t = 30$ minutes, LWP steadily increases up to a value of approximately $15.0 \, \text{g m}^{-2}$ at $t = 1$ hour (Fig. 2a).

At 1 hour into the simulation, ice processes are allowed to begin occurring within the model. This results in the rapid nucleation of virtually every IN-containing superdroplet within the cloud and a peak ice crystal number concentration of approximately $\sim 1.2 \, \text{L}^{-1}$ (Fig. 3a). The relatively high ice crystal number concentration and rapid growth of the ice crystals





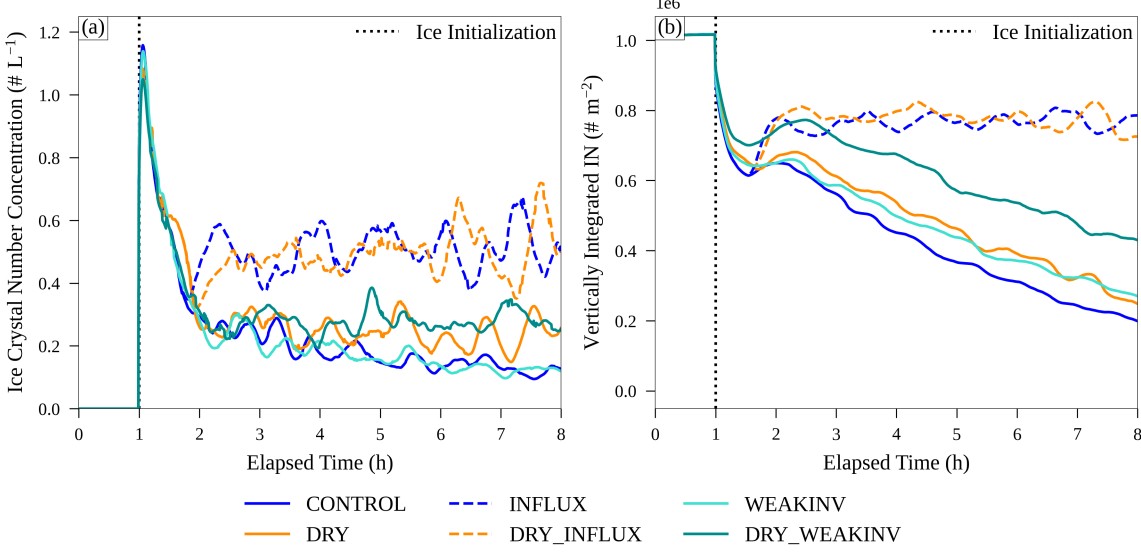

**Figure 3.** As in Figure 2, but showing ice crystal number concentration averaged within cloud (defined as regions with liquid water mixing ratio $> 0.01$ g kg$^{-1}$) (a) and vertically-integrated IN number concentration below cloud (b).

by vapor deposition leads to a classic WBF evolution and the net evaporation of cloud water, resulting in a decrease of LWP from $\sim$15.0 g m$^{-2}$ at $t$ = 1 hour to $\sim$12.6 g m$^{-2}$ at $t$ = 1.7 hours (Fig. 2a). Simultaneously, the IWP increases from 0.0 g m$^{-2}$ at $t$ = 1 hour to $\sim$7.6 g m$^{-2}$ at $t$ = 1.7 hours (Fig. 2b).

As the ice crystals grow larger, they begin to sediment out of the cloud, resulting in a rapid decrease in ice crystal number concentration within the cloud. Indeed, by $t$ = 2.3 hours, the average ice crystal number concentration within the cloud has plummeted to only $\sim$0.2 L$^{-1}$ (Fig. 3a). The depletion of ice crystal number concentration within the cloud is discussed in more detail below.

Though each individual ice crystal grows much more rapidly than the cloud droplets, the reduction in ice crystal number concentration means that the total rate of vapor deposition within the cloud decreases. So much so, in fact, that it decreases below the production of supersaturation through radiative and adiabatic cooling within the cloud, allowing the liquid cloud droplets to begin growing once more. After reaching a minimum at $t$ = 1.7 hours, the LWP steadily increases throughout the rest of the simulation, reaching $\sim$35.8 g m$^{-2}$ at $t$ = 8 hours (Fig. 2a).

This increase in cloud water is aided by a positive feedback loop within the cloud; liquid condensate near the cloud top emits infrared (longwave) radiation, leading to strong longwave cooling and the formation of a strong downdraft region of negatively buoyant air. This sinking air within this downdraft region, by conservation of momentum, leads to a surrounding region of ascent and supersaturation. Within this ascending, supersaturated region, even more liquid water content is produced, leading to greater longwave cooling at cloud top, and a stronger sinking motion within the downdraft (Petters et al., 2012). Cloud-top




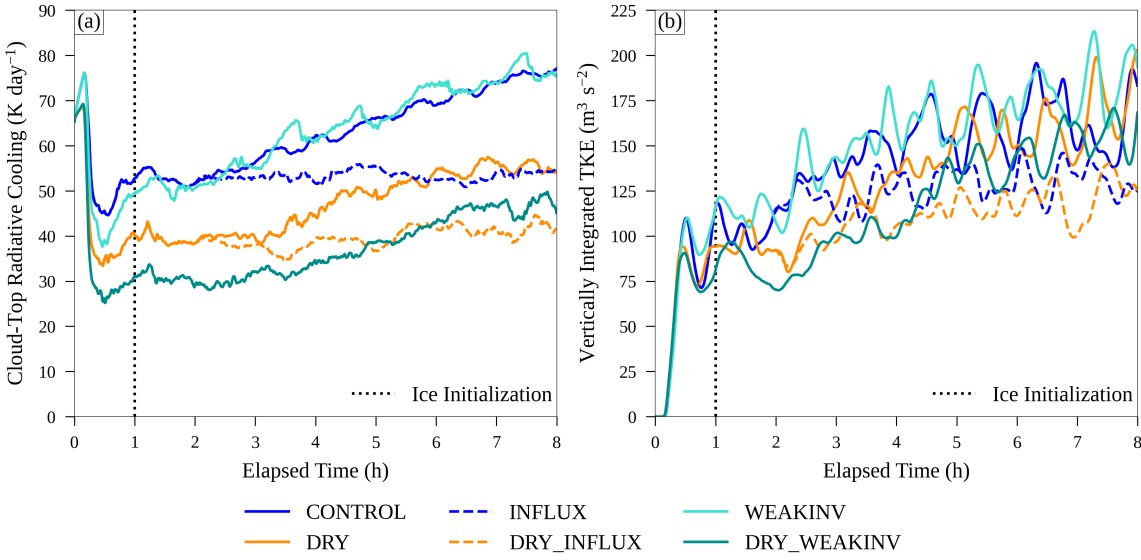

**Figure 4.** As in Figure 2, but displaying average radiative cooling rates within the uppermost 50 m of the cloud (a) and the vertically integrated turbulent kinetic energy up to cloud top (b).

radiative cooling rates increase from ∼55 K day$^{-1}$ at $t$ = 1.7 hours to ∼80 K day$^{-1}$ at $t$ = 8 hours (Fig 4a), while the vertically integrated Turbulent Kinetic Energy (TKE), hereafter referred to as the TKE path, increases from ∼80 m$^3$ s$^{-2}$ at $t$ = 1.7 hours to ∼160 m$^3$ s$^{-2}$ by $t$ = 8 hours (Fig. 4b). This increase in TKE path is a consequence not just of the strengthening of vertical motion and mixing within the cloud, but a deepening of the subcloud mixed layer. Importantly, because water vapor mixing ratios are higher in the decoupled near-surface layer, this means that the expanding below-cloud mixed layer entrains greater
and greater quantities of water vapor, also increasing the amount of liquid water within the cloud. At approximately 6.5 hours into the simulation, the mixed layer reaches the surface (not shown), meaning that the entire atmosphere up to the cloud top is well-mixed.

      The average size of ice crystals within the cloud increases throughout the simulation, from ∼500 $\mu$m at $t$ = 2 hours to ∼700 $\mu$m at $t$ = 8 hours (Fig. 5). The temperatures within the cloud fall between −13 °C and −15 °C, resulting in predominantly
plate-like growth of the crystals (Fig. S1). As mentioned briefly above, these large ice crystals fall through and below the cloud. Due to the lower equilibrium water vapor pressure over ice compared to liquid, the ice crystals continue to grow by vapor deposition even slightly below the cloud base, down to ∼450 m at $t$ = 1 hour. Below this altitude, however, the relative humidity with respect to ice falls below 100%, resulting in sublimation. This sublimation means that the number of ice particles reaching the surface is only ∼59.2% of those which are present at the lowest ice-supersaturated altitude, though this proportion
has considerable variability with time (Fig. S2).





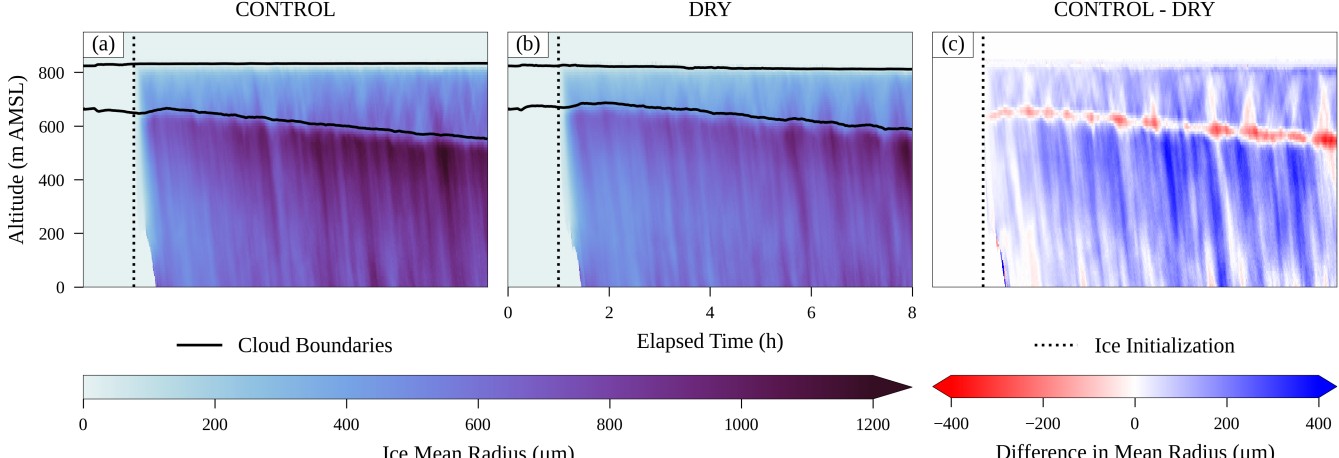

**Figure 5.** A time-height comparison of mean ice crystal radius (measured along the maximum particle dimension). (a,b) display radius in blue-purple shading, while (c) shows the difference in radius between CONTROL and DRY in red and blue colors. Cloud boundaries are shown in solid black lines in (a) and (b). Ice initiation at 1 hour is denoted by the vertical black dotted line.

This sublimation of ice crystals below cloud base is extremely important to the microphysical properties within the cloud itself. In the CONTROL simulation, there is no source of IN emission from the surface, while ice particles which reach the surface without sublimating are removed from the model domain, taking their IN with them. In contrast, ice particles which completely sublimate at some point below cloud base become dry IN. (We will refer to this complete sublimation, where a

particle's model classification changes from that of an ice crystal to that of a dry IN particle, as "desiccation".) This results in two effects: an overall depletion of IN from the domain, and a buildup of recycled IN below cloud base where ice crystals have desiccated.

A "band" of elevated IN concentrations below cloud base emerges approximately 1.5 hours into the simulation, initially at an altitude of ∼250 m. Note that this band does not form exactly where the majority of ice crystals desiccate, but rather

slightly below (Fig. 6). This is because the majority of ice crystals desiccate within the well-mixed region beneath cloud base, meaning that their IN are quickly spread throughout the mixed layer. The ice crystals which desiccate below the mixed layer, on the other hand, release their INs into a stable environment where they remain concentrated within a relatively narrow altitude range (the "band" visible in Fig. 6). As the mixed-layer deepens toward the surface, the altitude of this band sinks over time. At approximately 6.5 hours into the simulation, the "band" disappears as it intersects the surface. This indicates that there is no

longer a stable layer which desiccating ice particles can release their IN into. Instead, IN from desiccated ice crystals spread vertically throughout the mixed layer and into the cloud.

Nucleation rates at the cloud top are more than an order of magnitude less than those which occur at cloud base (Fig. 7a). Because the CONTROL simulation lacks an IN source at the surface, this means that the nucleation of new ice crystals within the cloud is thus almost entirely determined by the rate at which these recycled below-cloud IN are ingested into the cloud





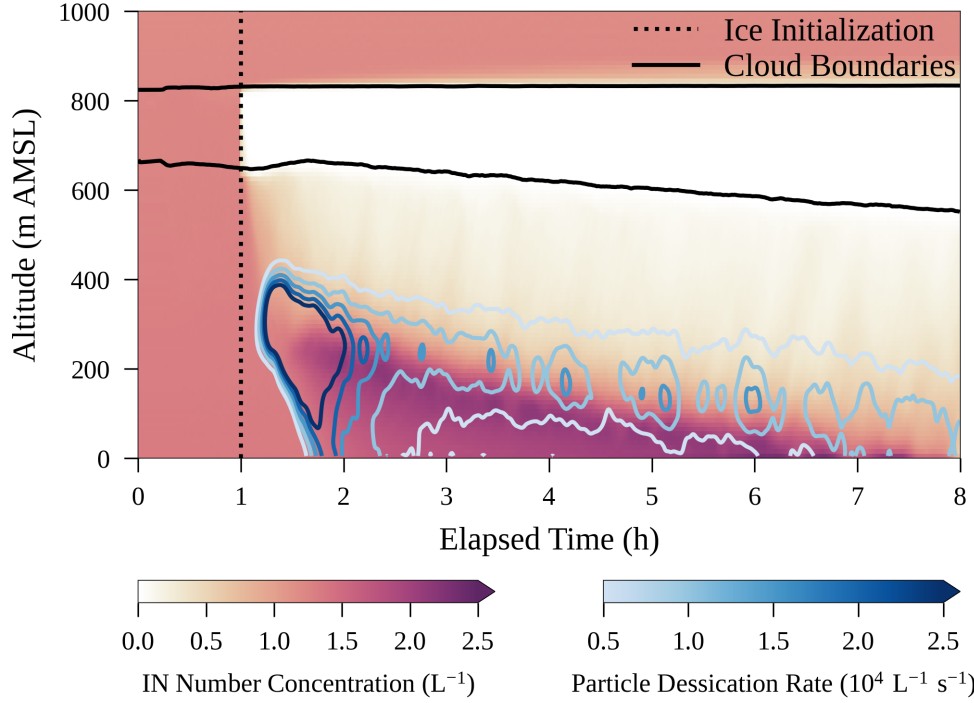

**Figure 6.** A time-height plot of IN number concentration (shaded) and ice crystal desiccation rate (in contours) in the CONTROL simulation. Contour values range from $5 \times 10^{-5}$ L$^{-1}$ s$^{-1}$ (light blue) to $2.5 \times 10^{-4}$ L$^{-1}$ s$^{-1}$ (dark blue). Cloud boundaries are indicated by solid black lines, while ice initiation at $t = 1$ hour is shown in the vertical black dotted line.

base. This leads to ice crystal nucleation and growth exclusively in updraft regions where recycled IN are lifted into the cloud from below, while downdraft regions are almost entirely liquid-phase (Figs. 7b, 8a,b). This total dependence of ice nucleation on recycled IN also means that the total number of IN in the domain decreases over time, as each round of ice nucleation, growth, sedimentation, sublimation, recycling, and re-nucleation always results in a loss of IN as some particles reach the surface without completely sublimating. Surprisingly, the actual number concentration of ice crystals within the cloud remains

in a roughly steady-state condition between 4 and 8 hours in the simulation. However, the mixed-phase nature of the cloud is moribund; with no way to replenish IN, the vertically-integrated IN concentration below cloud drops precipitously from $\sim 6 \times 10^5$ m$^{-2}$ at $t = 2$ hours to $\sim 2 \times 10^5$ m$^{-2}$ by $t = 8$ hours, continuing to fall rapidly as the simulation ends (Fig. 6). This indicates that the cloud, already liquid-dominated, will become almost entirely warm-phase with little ice or mixed-phase characteristics.

Another interesting consequence of heterogeneous ice nucleation occurring almost exclusively at cloud base is that ice formation and growth within the cloud occurs quite nonuniformly, colocated with updrafts which bring IN into liquid-saturated conditions. This means that ice within the cloud is almost entirely restricted to updrafts, with very little ice water content in



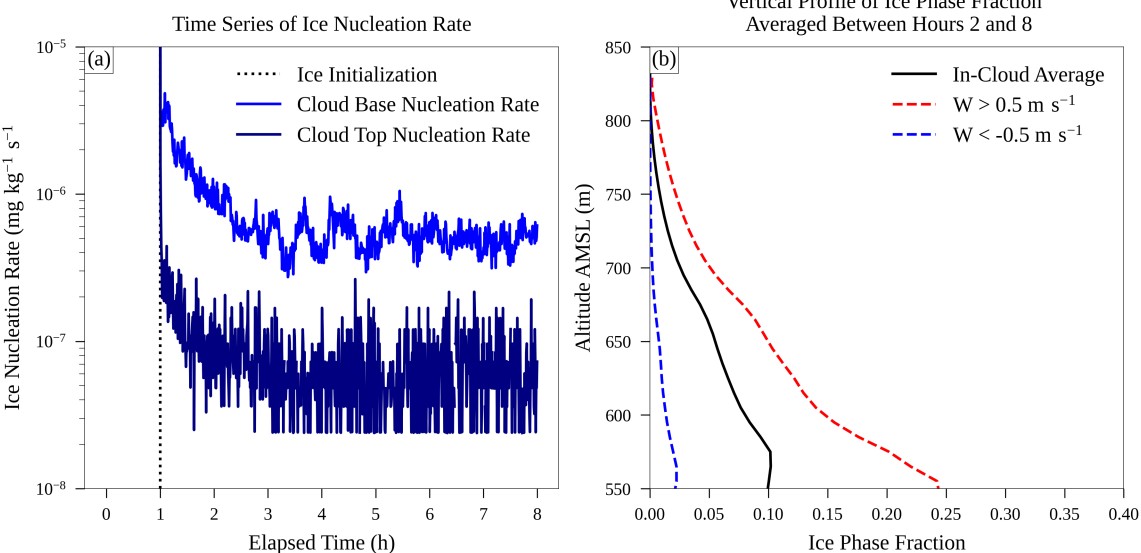

**Figure 7.** (a) A time series of heterogeneous ice nucleation rate at cloud base (blue) and cloud top (dark blue) in the CONTROL simulation. Ice initiation at $t$ = 1 hour is denoted by the black vertical dotted line. (b) A time-average vertical profile of ice phase fraction in updrafts (red, dashed), downdrafts(blue, dashed), and over the entire cloud (black, solid) between hours 2 and 8 in CONTROL.

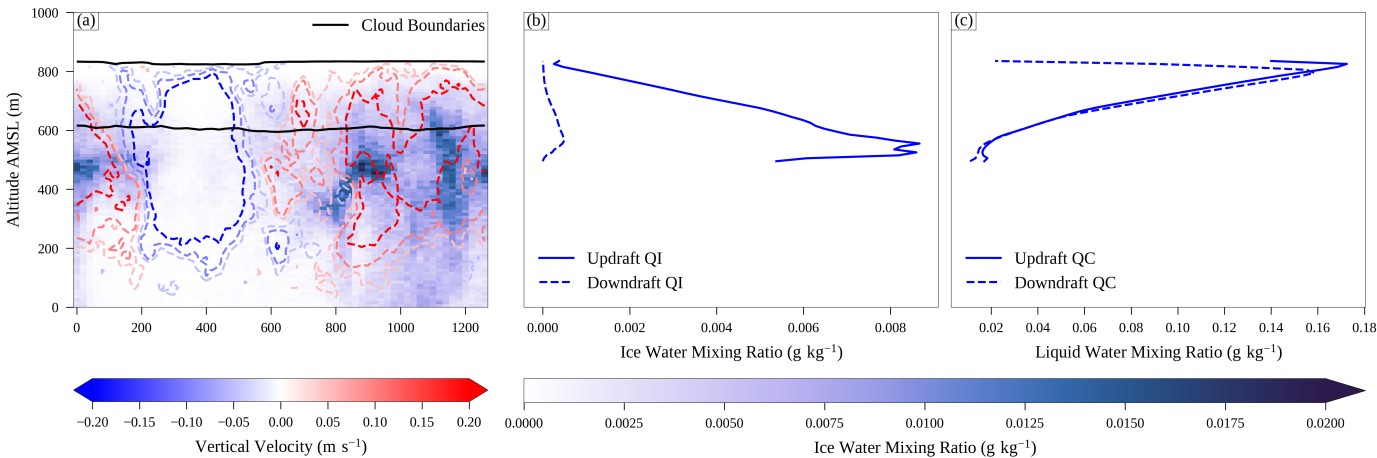

**Figure 8.** (a) A horizontal x-z cross-section of the CONTROL simulation at approximately 4.5 hours elapsed time. Ice water mixing ratio is denoted by blue shading, while contours of vertical velocity from $-0.2$ to $0.2$ m s$^{-1}$ are displayed in the blue and red contours (for downdrafts and updrafts, respectively). Cloud boundaries are denoted by solid black lines. (b) Time-average vertical profile of ice water mixing ratio in updrafts (solid) and downdrafts (dashed) between hours 2 and 8. (c) same as (b), but for liquid water mixing ratio.



downdraft regions, similar to the modeling results of Raatikainen et al. (2022) (Figs. 7, 8a,b). The almost complete confinement of ice particles to updraft regions means that there is essentially no ice growth by vapor deposition in downdraft regions. Since

downdraft regions are entirely liquid, with no competition from ice, this leads to the rather surprising result that downdraft regions of the cloud actually have a slightly higher liquid water content than updraft regions (Fig. 8c)!

## 3.2  Sensitivity to Humidity - DRY

The DRY simulation displays greatly reduced LWPs and cloud depths compared to CONTROL. LWP just before the onset of ice formation at $t = 1$ hour is only $\sim$10.2 g m$^{-2}$, compared to $\sim$15.0 g m$^{-2}$ in CONTROL (Fig. 1a). This reduction in

LWP compared to CONTROL persists and widens as the simulation progresses. Although LWP does still increase steadily in DRY from hours 2 to 8, it does so at a slower rate than in CONTROL. Between hours 2 and 8, the average LWP in DRY is $\sim$14.3 g m$^{-2}$, $\sim$40.6% less than the average of $\sim$24.1 g m$^{-2}$ in CONTROL (Fig. 1a). This reduction in LWP is due to the entrainment of drier above-cloud air than in CONTROL. As discussed in Section 2, the initial water vapor mixing ratio above cloud is set to 0.8 g kg$^{-1}$ in DRY, compared to 1.2 g kg$^{-1}$ in CONTROL. This corresponds to an initial above-cloud relative

humidity of $\sim$41.3% in DRY and $\sim$61.9% in CONTROL.

This reduction in LWP also slows down (but does not eliminate) the positive feedback loop which deepens the cloud in CONTROL. With a lower initial LWP, radiative cooling at cloud top is reduced in DRY compared to CONTROL. Between hours 2 and 8, the average longwave radiative cooling rates within the top 50 meters of the cloud are $\sim$65.2 K day$^{-1}$ and $\sim$48.6 K day$^{-1}$ in CONTROL and DRY, respectively (Fig. 4a).

However, the most interesting aspect of DRY is the evolution of ice within and below the cloud. The initial "pulse" of ice formation which begins at $t = 1$ hour and continues until approximately 2 hours is weaker in DRY than CONTROL (Fig. 2b). Even after the "pulse", IWP remains depressed in the DRY simulation compared to the CONTROL simulation until approximately $t = 4.75$ hours, after which IWP increases above the value in CONTROL (Fig. 2b).

These results were unexpected, and, at first glance, seem to corroborate those of Hoffmann (2020), that entrainment of dry

air and evaporation of cloud droplets lead to an enhancement of ice crystal growth through the WBF process. However, this is not in fact the case in our experiments; rather, the increase in IWP is due to differences in IN recycling between the CONTROL and DRY simulations. As discussed above, recycling of IN below cloud base by the desiccation of sedimenting ice crystals is the only way in which the number concentration of below-cloud IN can be increased. Without this recycling, IN which were consumed by nucleation would always be removed from the domain as ice crystals reached the surface, quickly starving the

simulation of IN and leading to a purely liquid cloud. This therefore means that the amount of recycled IN is therefore a critical factor in controlling the rate of new ice nucleation and growth within the cloud and a major determinant of total IWP.

In DRY, two important factors lead to much more efficient IN recycling than in CONTROL. The first is that ice crystals simply don't experience as much vapor depositional growth in DRY as they do in CONTROL. Per-particle deposition rates are reduced in DRY compared to CONTROL, as is the vertical depth of ice-supersaturated air (Fig. 9). As a result, ice crystals

in DRY experience reduced maximum growth rates and spend less time within ice-supersaturated conditions than those in CONTROL. This leads to smaller ice crystals when they begin to sublimate. At the altitude corresponding to 100% RH with



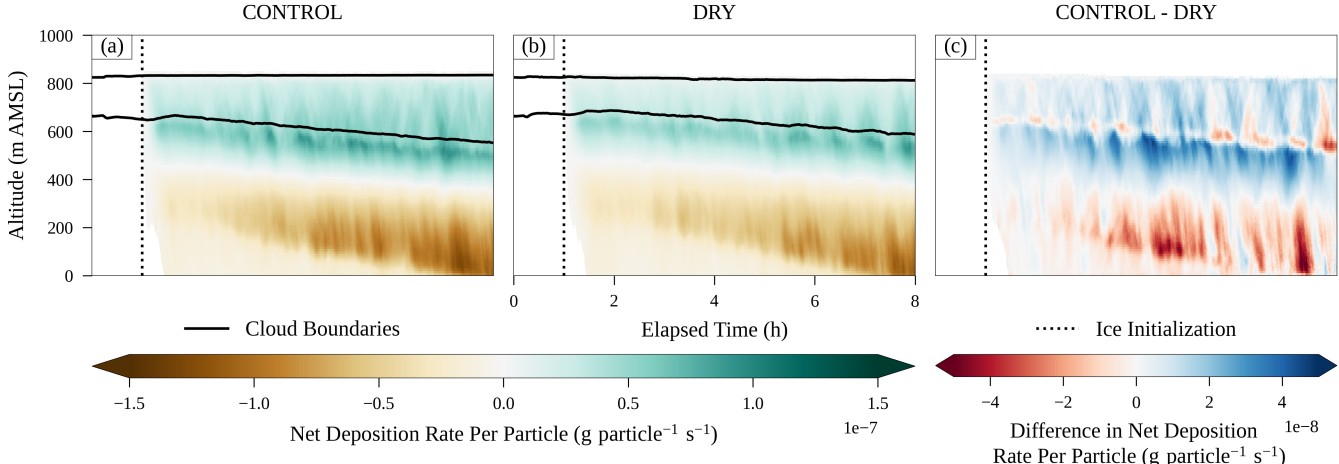

**Figure 9.** A time-height comparison of net deposition rate per particle. Deposition rates per particle are shown in green and brown shading in (a,b), while differences between CONTROL and DRY are shown in blue and red shading in (c). Cloud contours for respective simulations are indicated by solid black lines in (a,b), while the initiation of ice processes at $t = 1$ hour is shown by the vertical dotted black lines.

respect to ice (below which falling ice crystals will sublimate), the mean ice radius (averaged over the long axis of each particle) is ~872 $\mu$m in CONTROL and only ~705 $\mu$m in DRY (Figs. 5, S3).

Compounding the fact that ice crystals in DRY are smaller when they begin to sublimate, the below-cloud mixed layer is
also drier in DRY. Between hours 2 and 8, the average relative humidity over ice in the ice-subsaturated region below cloud is ~91.6% in DRY, compared with ~92.4% in CONTROL. Compounding the greater dryness is the fact that the depth of this subsaturated layer is also approximately 30 meters deeper in the DRY compared to CONTROL (Fig. 10a).

Together, the combination of the smaller size of ice crystals when they begin to sublimate, and the deeper and drier layer in which they sublimate, leads to a greater fraction of ice crystals in DRY desiccating rather than reaching the surface. Indeed,
between hours 2 and 8, the average number concentration of ice crystals at the lowest model level in DRY is only ~37.5% of the number concentration at the lowest ice-supersaturated altitude (not shown). By contrast, ~59.2% of sedimenting ice crystals at the lowest ice-supersaturated altitude reach the surface in CONTROL (Fig. S2b). This greater rate of desiccation in DRY in turn increases the recycling of IN below cloud (Figs 3b, 11). Since ice crystal number concentrations within the cloud are primarily determined by the rate of uptake of recycled IN from below the cloud, DRY, with higher recycled IN concentrations,
has much higher in-cloud ice crystal number concentrations than CONTROL (Fig. 3a). Although each individual ice crystal is, on average, smaller in DRY, the fact that there are more of them means that the total IWP eventually becomes larger than in CONTROL (Figs 2b, 5, S3). This suggests that the increase in IWP in DRY is driven by differences in ice crystal number concentration, rather than any manifestation of entrainment-enhanced ice growth as reported in Hoffmann (2020). The effects of IN recycling on ice crystal number concentration and IWP are discussed further in Sections 3.5 and 3.6.




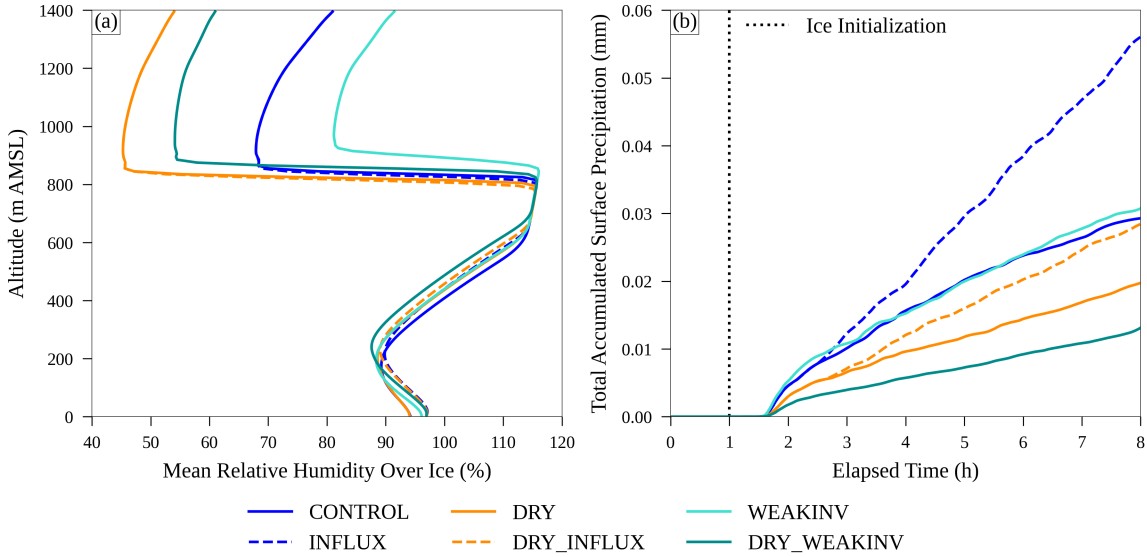

**Figure 10.** (a) Profiles of relative humidity with respect to ice averaged over time between hours 2 and 8. (b) Time series of surface precipitation (in mm of water equivalent) in six simulations. Ice initiation at $t = 1$ hour is indicated by the vertical dotted black line in (b).

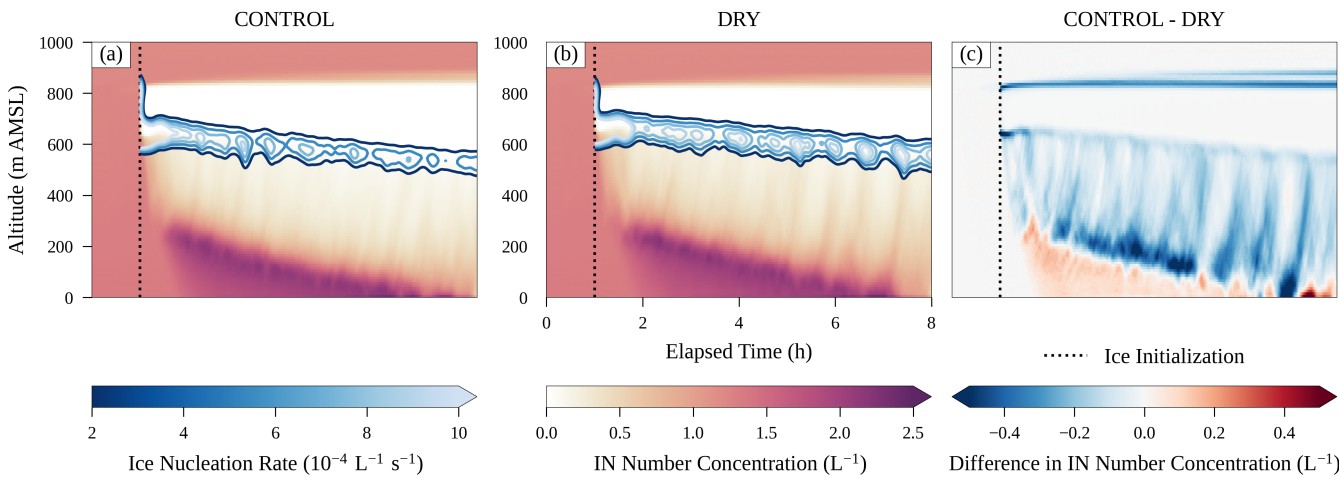

**Figure 11.** A time-height comparison of IN number concentration and nucleation rate. (a,b) show IN number concentration in pink-purple shading and ice nucleation rate in blue contours. The contour range is from $2 \times 10^{-4}$ particle $L^{-1}$ to $1 \times 10^{-3}$ particle $L^{-1}$ with a contour interval of $2 \times 10^{-4}$ particle $L^{-1}$. Contours are smoothed in time with a gaussian kernel with a 2.5 minute standard deviation for clarity. (c,f) show the difference in IN number concentration between CONTROL and DRY in red and blue shading.



## 3.3 Sensitivity to Inversion Strength - WEAKINV

Cloud top height increases steadily over the course of the WEAKINV simulation, unlike CONTROL, in which cloud top height remains almost constant. This is, unsurprisingly, due to an increase in the entrainment rate at cloud top in WEAKINV compared to CONTROL. The average entrainment rate between hours 2 and 8 in CONTROL is $\sim$4.20 mm s$^{-1}$, just barely larger than the prescribed large-scale subsidence velocity of 4.12 mm s$^{-1}$ (Fig. S4a). By contrast, the average entrainment rate in WEAKINV is $\sim$54.4% higher, at $\sim$6.48 mm s$^{-1}$. While cloud base height lowers over time in both simulations (as the cloud deepens), it does so at a slower rate in WEAKINV than CONTROL. Cloud base height sinks at an average rate of $\sim$5.03 mm s$^{-1}$ in CONTROL, compared to only $\sim$3.30 mm s$^{-1}$ in WEAKINV (not shown). This slower rate of cloud base decrease in WEAKINV is due to the greater entrainment of warm air from the entrainment interfacial layer (EIL) into and beneath the cloud, which raises the temperature of the boundary layer compared to CONTROL. This warming of the boundary layer in WEAKINV also raises the height of the cloud base compared to CONTROL. Together, the combination of higher cloud base and higher cloud top in WEAKINV mean that the average cloud thickness between hours 2 and 8 is $\sim$249 m, a $\sim$8.35% increase compared to the average cloud thickness of $\sim$229 m in CONTROL (Fig. S4b).

The deeper cloud in WEAKINV leads to a larger LWP compared to CONTROL, with an average value of $\sim$27.4 g m$^{-2}$ between hours 2 and 8, a 13.5% increase over the average LWP of $\sim$24.2 g m$^{-2}$ in CONTROL (Fig. 2a). This increase in LWP is due to a combination of thicker cloud and greater condensate mixing ratio compared to CONTROL. As discussed previously, the cloud thickness is $\sim$8.35% greater in WEAKINV than CONTROL. At the same time, average in-cloud liquid water mixing ratio is $\sim$5.2% greater in WEAKINV compared to CONTROL (not shown). This is primarily because the greater cloud depth in WEAKINV results in a deeper region of moist adiabatic ascent, and therefore greater total supersaturation production, compared to CONTROL. Though one would expect the greater entrainment rate in WEAKINV to mix more dry air into the cloud, slowing the rate of condensation, this does not occur to any great extent. Indeed, the average condensational growth rate per particle is only $\sim$0.85% less in WEAKINV than it is in CONTROL (not shown). The relative lack of drying in WEAKINV, despite the greater entrainment rate, is due to the fact that the above-cloud air in WEAKINV is at a higher relative humidity than that in CONTROL, as a consequence of reducing the inversion strength while leaving the above-cloud mixing ratio unchanged. The average relative humidity of air 50 meters above cloud top (above the entrainment interfacial layer) between hours 2 and 8 in CONTROL is $\sim$61.9%, considerably drier than the $\sim$75.8% average relative humidity of above-cloud air in WEAKINV. Though not shown directly, differences in above-cloud relative humidity between simulations can be inferred from Fig. 10a.

The greater cloud depth in WEAKINV also increases the lifetime of ice crystals ascending and falling through the cloud compared to CONTROL. Between hours 2 and 8, the average in-cloud lifetime of ice crystals in WEAKINV is $\sim$4.5% longer than that in CONTROL (not shown). Coupled with a per-particle deposition rate which is $\sim$7.2% greater in WEAKINV than CONTROL, this leads to an $\sim$11.2% increase in average in-cloud ice water mixing ratio between hours 2 and 8 in WEAKINV compared to CONTROL. This increased deposition rate in WEAKINV is the result of multiple factors, including a greater average vertical velocity variance in WEAKINV and greater lifetime of ice crystals. The greater vertical velocity variance in



WEAKINV translates to higher updraft (and downdraft) speeds, resulting in a greater production of supersaturation compared to CONTROL. The greater lifetime of ice, meanwhile, also results in greater average per-particle deposition rates as older and larger ice particles experience faster depositional growth. The greater depositional growth rate and deeper cloud depth in WEAKINV means that the average IWP between hours 2 and 8 is $\sim$12.1% greater than that in CONTROL, at $\sim$3.53 g m$^{-2}$ compared to $\sim$3.15 g m$^{-2}$ in CONTROL (Fig. 2b). Ice crystals at the lowest ice-supersaturated altitude are also larger in WEAKINV than CONTROL, with an average major-axis diameter of $\sim$913 $\mu$m compared to $\sim$871 $\mu$m in CONTROL (Fig. S3).

The greater entrainment rate of free-tropospheric air in WEAKINV has important consequences for the thermodynamics of the sub-cloud boundary layer. The ingestion of warm air downward into the cloud results in a higher cloud base in WEAKINV compared to CONTROL and a warmer, drier sub-cloud boundary layer. The average relative humidity with respect to ice throughout the sub-cloud boundary layer is $\sim$1.08% smaller in WEAKINV compared to CONTROL (Fig. 10a). Interestingly, however, the fraction of ice crystals present at the lowest ice-supersatured altitude in WEAKINV which reach the surface is $\sim$57.5%, barely different from the average value of $\sim$59.3% in CONTROL. Similarly, both CONTROL and WEAKINV show similar amounts of ice precipitation at the surface (Fig. 10b). This suggests that the greater ice crystal size in WEAKINV and the drier sub-cloud boundary layer compared to CONTROL have counteracting effects on ice crystal desiccation rate, leading to a similar overall fraction of ice crystals which desiccate.

## 3.4 Combining Humidity and Inversion Strength - DRY_WEAKINV

DRY_WEAKINV displays the lowest LWP of any of the four sensitivity simulations. Average LWP between hours 2 and 8 in DRY_WEAKINV is only $\sim$10.5 g m$^{-2}$, a reduction of $\sim$56.4%, $\sim$26.6%, and $\sim$61.6% from CONTROL, DRY, and WEAKINV, respectively (Fig. 2a). DRY_WEAKINV also has the thinnest cloud of the 4 simulations; the average cloud depth of $\sim$160.7 m between hours 2 and 8 is a $\sim$30% decrease compared to CONTROL, a $\sim$11.2% decrease compared to DRY, and a $\sim$35.4% decrease from WEAKINV (Fig. S4b). The fact that the cloud in DRY_WEAKINV is thinner than that in DRY is particularly surprising given the fact that the cloud top height increases slowly throughout the course of the simulation in DRY_WEAKINV while it falls slowly during DRY (not shown). However, the cloud base in DRY_WEAKINV is the highest of all simulations. These results present a sharp contrast with WEAKINV, which displayed a higher LWP and deeper cloud than CONTROL. The differences between WEAKINV and DRY_WEAKINV clearly illustrates that not just the magnitude, but also the sign of the impacts of entrainment on cloud microphysical properties is strongly affected by the relative humidity of the free troposphere.

The reduction in LWP in DRY_WEAKINV inhibits the rate of longwave cooling at cloud top. Between hours 2 and 8, the average longwave cooling rate within the uppermost 50 meters of the cloud is $\sim$38.4 K day$^{-1}$, a $\sim$41.1%, $\sim$21.0%, and $\sim$42.1% reduction compared to CONTROL, DRY, and WEAKINV, respectively (Fig. 4a). However, even in DRY_WEAKINV, this reduced longwave cooling rate is sufficient to promote a positive feedback loop of cloud developmeent as in CONTROL, DRY, and WEAKINV. Longwave cooling rates increase from approximately $\sim$29.1 K day$^{-1}$ at hour 2 to $\sim$45.2 K day$^{-1}$ at hour 8, as LWP likewise increases from $\sim$6.2 g m$^{-2}$ to $\sim$16.3 g m$^{-2}$ over the same time period. This reduced longwave




cooling also suppresses turbulence in DRY_WEAKINV compared to the other simulations, with a TKE path ∼18.6%, ∼11.9%, and ∼23.8% smaller than those in CONTROL, DRY, and WEAKINV, respectively (Fig. 4b). This reduction in longwave

cooling and TKE path may act to limit cloud-top entrainment; mean cloud-top entrainment rate between hours 2 and 8 in DRY_WEAKINV is ∼4.74 $\mathrm{mm\ s^{-1}}$, a ∼26.7% decrease compared to CONTROL (Fig. S4a). This suggests the presence of a negative feedback in the presence of weak inversion strength and dry above-cloud air, in which initial dry air entrainment decreases liquid water content and radiative cooling at cloud top, reducing turbulence and cloud-top entrainment, helping to insulate the cloud from further drying.

IWP in DRY_WEAKINV, as in all simulations, is highly variable over time. The initial "pulse" of ice formation is strongly reduced compared to DRY, CONTROL, or WEAKINV, with a peak IWP of 4.9 $\mathrm{g\ m^{-2}}$ (Fig. 2b). This constitutes a ∼35.9%, ∼24.6%, and ∼45.0% reduction compared to CONTROL, DRY, and WEAKINV, respectively. Between hours 2 and 8, IWP in DRY_WEAKINV is ∼3.7%, ∼7.0%, and ∼14.1% reduced compared to CONTROL, DRY, and WEAKINV, respectively. However, this average difference masks the fact that IWP in DRY_WEAKINV increases over time relative to the other three

simulations (Fig. 2b). As with DRY, this relative increase in IWP is due to the more efficient recycling of IN beneath the cloud compared to the CONTROL or WEAKINV simulations. Examining Figure 3b, the DRY_WEAKINV simulation stands out as having the highest below-cloud IN quantities of any simulation without IN re-injection.

The very efficient IN recycling in DRY_WEAKINV is due to the same factors as those which led to greater IN recycling in DRY: reduced ice crystal depositional growth rates resulting in smaller ice crystals, and a deep and dry ice-subsaturated layer.

Between hours 2 and 8, the average depth of the ice-subsaturated layer below cloud is ∼493 m in DRY_WEAKINV, compared to ∼414 m, ∼442 m, and ∼445 m in CONTROL, DRY, and WEAKINV, respectively. Simultaneously, the average relative humidity with respect to ice in the subsaturated region of the sub-cloud boundary layer in DRY_WEAKINV is ∼2.8%, ∼1.2%, and ∼1.3% less than that in CONTROL, DRY, and WEAKINV, respectively (Fig. 10a). Ice crystals sedimenting into the ice-subsaturated layer are also smaller than those in the other three simulations, with a major axis diameter of only ∼625 $\mu$m

compared to ∼871 $\mu$m, ∼705 $\mu$m, and ∼913 $\mu$m in CONTROL, DRY, and WEAKINV, respectively (Fig. S3). Taken together, it's therefore unsurprising that a greater fraction of ice crystals in DRY_WEAKINV desiccate before reaching the surface than in the other three sensitivity simulations. In fact, between hours 2 and 8, only ∼29.7% of ice crystals present at the lowest ice-supersaturated altitude in DRY_WEAKINV reach the surface, compared to ∼59.3% in CONTROL, ∼37.5% in DRY, and ∼57.5% in WEAKINV (not shown). Due to both fewer and smaller ice crystals reaching the surface in DRY_WEAKINV com-

pared to the other three simulations, ice precipitation at the surface is also lower in DRY_WEAKINV compared to CONTROL, DRY, or WEAKINV (Fig. 10b).

### 3.5 Sensitivity to IN - INFLUX

INFLUX has much-reduced cloud condensate compared to CONTROL, particularly in the latter half of the simulation. Unlike in CONTROL, where both the cloud depth and LWP steadily increase between hours 2 and 8, in INFLUX these quantities both

remain fairly stable with time. Between hours 2 and 8, the average LWP in INFLUX is ∼13.5 $\mathrm{g\ m^{-2}}$, ∼44.0% less than the average value of ∼24.1 $\mathrm{g\ m^{-2}}$ in CONTROL (Fig. 2a). Interestingly, this is a similar magnitude of LWP reduction as seen in





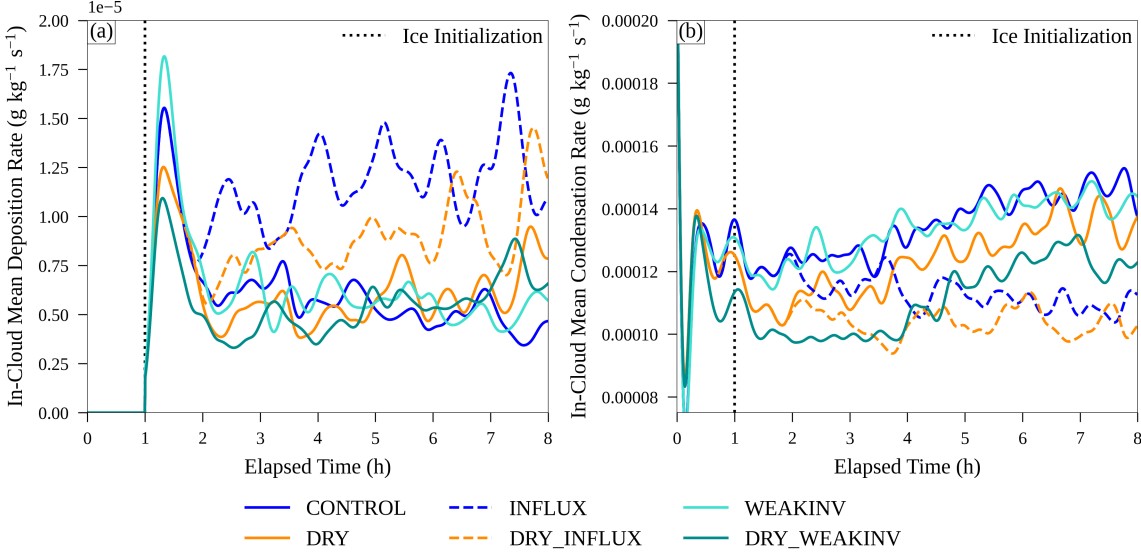

**Figure 12.** Time series of average in-cloud deposition rate (a) and condensation rate (b). Note that this is not the deposition rate per particle, but rather per mass of air. Ice initiation at $t = 1$ hour is indicated by the vertical dotted black lines. Quantities are smoothed with a gaussian time filter with a standard deviation of 5 minutes for clarity.

DRY. For the same time period, average IWP in INFLUX is ~102% higher than that in CONTROL, at ~6.36 g m$^{-2}$ compared to ~3.15 g m$^{-2}$ (Fig. 2b).

Unsurprisingly, the reduction in LWP and increase in IWP compared to CONTROL are a result of the greater number of
365 ice crystals in INFLUX depleting more water vapor from the cloud through vapor deposition. Though deposition rates per ice particle are higher in CONTROL simulation compared to INFLUX, the much greater ice crystal number concentration in INFLUX simulation leads to greater total deposition rates. In INFLUX, the average vapor deposition rate per particle between hours 2 and 8 is ~25.4 ng particle$^{-1}$ s$^{-1}$, a ~27.9% decrease from the average value of ~35.2 ng particle$^{-1}$ s$^{-1}$ in CONTROL (not shown). However, because the average in-cloud ice crystal number concentration at the same time is ~197%
higher in INFLUX than CONTROL, at ~0.51 particles L$^{-1}$ compared to ~0.17 particles L$^{-1}$, the total in-cloud deposition rate in INFLUX is $1.18 \times 10^{-5}$g kg$^{-1}$ s$^{-1}$, ~121% higher than the average value of $5.35 \times 10^{-6}$g kg$^{-1}$ s$^{-1}$ in CONTROL (Figs. 3 12).

This greater consumption of water vapor through vapor deposition on ice crystals in INFLUX thus reduces the amount of condensational growth compared to CONTROL. Between hours 2 and 8, average in-cloud condensation rates in INFLUX are
375 ~81.6% of those in CONTROL, at $1.12 \times 10^{-4}$g kg$^{-1}$ s$^{-1}$ compared to $1.38 \times 10^{-4}$g kg$^{-1}$ s$^{-1}$ (Fig. 12). Comparing only condensation rates actually understates the difference between the two simulations, since, as can be seen in Fig. S5, not only are condensation rates higher in CONTROL than in INFLUX, but they occur over a greater depth as the cloud is thicker. In fact, while the cloud continuously grows deeper in both CONTROL and DRY, it remains at an almost constant thickness in





INFLUX. This suggests that the feedback loops present in the CONTROL and DRY simulations between cloud condensate, radiative cooling, production of turbulence, and supersaturation production may be inhibited with certain concentrations of IN, in line with previous research finding that large enough IN concentrations will cause complete cloud glaciation and/or dissipation (Das et al., 2025; Loewe et al., 2017; Morrison et al., 2005; Paukert and Hoose, 2014). Interestingly, the results from INFLUX disagree quantitatively with those of Ovchinnikov et al. (2014), who found that a background IN concentration of $1\,\mathrm{L}^{-1}$ supported a steady increase in LWP with time from $\sim 10\,\mathrm{g\,m}^{-2}$ to between 40 and $50\,\mathrm{g\,m}^{-2}$. Our INFLUX simulation, by contrast, agrees better with their simulations using fixed IN concentrations of $4\,\mathrm{L}^{-1}$, in which LWP changes little with time, remaining between 10 and $20\,\mathrm{g\,m}^{-2}$. This may be because Ovchinnikov et al. (2014) utilized an ice crystal growth parameterization with a specified mass-diameter relationship which did not allow for variable habit development as in our model.

The reduction in LWP in the INFLUX simulation, similarly to the DRY simulation, acts to inhibit cloud-top longwave radiative cooling compared to CONTROL. Average longwave cooling rates between hours 2 and 8 in the uppermost 50 meters of the cloud are $\sim 53.3\,\mathrm{K\,day}^{-1}$ in INFLUX, $\sim 18.3\%$ less than the $\sim 65.2\,\mathrm{K\,day}^{-1}$ in CONTROL (Fig. 4a). Coupled with the decrease in condensational latent heating, this leads the INFLUX simulation to have an average TKE path between hours 2 and 8 which is $\sim 17.1\%$ smaller than that in CONTROL (Fig. 4).

### 3.6 Combining Humidity and IN - DRY_INFLUX

The DRY_INFLUX simulation makes for a very interesting comparison, allowing us to analyze how the interaction of thermodynamic and IN modification affect the development of the cloud and boundary layer differently than either modification in isolation. The DRY_INFLUX has the lowest LWP or cloud depth compared to either CONTROL, DRY, or INFLUX. Indeed, its average LWP between hours 2 and 8 is a mere $8.8\,\mathrm{g\,m}^{-2}$, 38.3% less than in DRY, 34.6% less than in INFLUX, and 63.4% less than CONTROL (Fig. 2). This is because water vapor within the cloud is depleted both by the enhanced vapor deposition rates from higher ice crystal number concentrations and by increased evaporation through mixing with the dry air above the cloud. As discussed above in Section 3.5, the decrease in LWP associated with higher ice crystal number concentrations in the INFLUX simulation is of a similar magnitude to that associated with enhanced evaporation in the DRY simulation, when compared to LWP in CONTROL. This suggests that, under the environmental conditions of the ISDAC simulation, the effects of perfectly efficient IN recycling and the magnitude of the mixing ratio decrease in DRY cause a similar magnitude of supersaturation depletion within the cloud.

This reduction in LWP also leads to a large reduction in longwave radiative cooling in the DRY_INFLUX simulation. Between hours 2 and 8, the average cooling rate in the highest 50 meters of the cloud is only $39.6\,\mathrm{K\,day}^{-1}$, compared with $65.2\,\mathrm{K\,day}^{-1}$ in CONTROL, a 39.3% reduction. This is also 28.2% less than INFLUX and 18.4% less than DRY (Fig. 4a). This decrease in cloud-top radiative cooling results in a 26.6% reduction of TKE path from $153\,\mathrm{m}^3\,\mathrm{s}^{-2}$ in CONTROL to $112\,\mathrm{m}^3\,\mathrm{s}^{-2}$ in DRY_INFLUX (TKE path in DRY_INFLUX is also 12.1% lower than that in INFLUX and 21.1% less than in DRY) (Fig. 4b). However, the cloud appears to be in no danger of glaciating or dissipating in DRY_INFLUX, as LWP increases slowly but steadily from $8\,\mathrm{g\,m}^{-2}$ at hour 4 to $9.5\,\mathrm{g\,m}^{-2}$ at hour 8 (Fig. 2a).



IWP in DRY_INFLUX is lower than that in INFLUX, while higher than that in either DRY or CONTROL. Between hours 2 and 8, the average IWP in DRY_INFLUX is 4.55 $\mathrm{g\,m^{-2}}$, 44.6% and 39.6% higher than CONTROL and DRY, respectively, while also reduced by 28.5% compared with INFLUX (Fig. 2b). Similar to our comparison of INFLUX and CONTROL, in-cloud deposition rates per particle in DRY_INFLUX are smaller than those in the CONTROL or DRY simulations, at approximately 20.1 $\mathrm{ng\,particle^{-1}\,s^{-1}}$ in DRY_INFLUX, compared to 35.2 $\mathrm{ng\,particle^{-1}\,s^{-1}}$ in CONTROL and 26.6 $\mathrm{ng\,particle^{-1}\,s^{-1}}$ in DRY (not shown). Notably, the average ice radii are smaller both within and below cloud base in the DRY_INFLUX simulation compared with CONTROL, DRY, or INFLUX; Average ice particle radius at the lowest ice-supersaturated altitude is approximately 610 $\mu$m, compared to 728 $\mu$m in INFLUX, 705 $\mu$m in DRY, and 872 $\mu$m in CONTROL (Fig. S3). However, the much greater ice crystal number concentration in DRY_INFLUX means that both the total deposition rate and the IWP are higher than those in either CONTROL or DRY (Fig. 12a).

As discussed above in the comparison of DRY and CONTROL, the smaller size of ice crystals when they begin to sublimate and the drier sub-cloud sublimation layer within DRY_INFLUX both contribute to increased desiccation rates of ice crystals and a reduction in the number and mass of ice crystals reaching the surface compared to CONTROL. This is most strikingly seen in a comparison of accumulated precipitation over the course of the six simulations, which illustrates that, despite the much greater IWP in DRY_INFLUX compared to CONTROL, the total accumulated snowfall at the surface is higher in CONTROL throughout almost the entire simulation is higher than in DRY_INFLUX, due to higher sub-cloud sublimation rates in the latter simulation (though, by the end of the simulation, CONTROL has become so IN-depleted that it is likely that running both simulations for another 8 hours would result in DRY_INFLUX overtaking CONTROL in terms of total surface precipitation) (Fig. 10b). Unlike DRY and CONTROL, however, the increased sublimation and desiccation rates within DRY_INFLUX result neither in a higher IN number concentration below cloud nor a higher in-cloud ice crystal number concentration compared to INFLUX, as the total amount of IN-containing particles (both "dry" INs and ice crystals containing INs) are held constant in both simulations.

## 4 Discussion and Conclusions

To examine the sensitivity of Arctic mixed-phase clouds to environmental humidity, inversion strength, and IN depletion, we ran six simulations using an LES model with Lagrangian cloud microphysics. Our results demonstrate the clear feedback between the microphysical and boundary-layer processes in Arctic mixed-phase stratocumulus clouds, and the role of ice in determining this feedback.

Reduced above-cloud relative humidity in DRY results in both expected and unexpected impacts on cloud and boundary-layer properties. Due to increased cloud-top evaporation, LWP within the cloud is reduced compared to CONTROL, as is, initially, IWP. By the end of the DRY simulation, however, IWP is higher than in the CONTROL simulation, though LWP remains much lower (Fig. 2). This is due to enhanced IN recycling driven by desiccation of sublimating ice crystals. The entrainment of drier above-cloud air into the cloud reduces the available supersaturation for ice crystal growth by vapor deposition, resulting in smaller ice crystals (Fig. 5). Simultaneously, turbulent transport of this drier above-cloud air through



and below the cloud reduces water vapor mixing ratios below cloud base, increasing the water vapor deficit experienced by ice crystals below the cloud. Both of these factors result in a greater number of sedimenting ice crystals desiccating before reaching the surface and returning their IN to the boundary layer, which are then nucleated once more at cloud base (Fig. 11). The higher number concentration of ice crystals in the DRY simulation compared to CONTROL therefore results in a higher

final IWP, even though the average size of each individual ice crystal is smaller in DRY than CONTROL.

We do not find any evidence to support the hypothesis of dry air entrainment enhancing ice crystal growth, as hypothesized in Hoffmann (2020). The increased IWP present in the DRY simulation would seem to lend credence to this hypothesis, but, as described above, the increase in IWP in the DRY simulation is solely a result of more efficient IN recycling. Within the DRY simulation, deposition rates per particle are reduced compared to CONTROL, but the greater number concentration of

455 ice crystals within cloud results in a higher total deposition rate and IWP (Figs. 9, 12a). Examination of the INFLUX and DRY_INFLUX simulations, both of which run with perfectly efficient IN recycling, clearly shows a reduction in IWP in the latter simulation (Fig. 2b). Depositional growth rates, both per particle and total, are lower in DRY_INFLUX than INFLUX, as entrainment of dry air reduces the available supersaturation for ice crystal growth.

The WEAKINV and DRY_WEAKINV simulations support the finding of several previous studies of subtropical stratocu-

460 mulus that above-cloud relative humidity plays a critical role in modulating the effects of cloud-top entrainment on cloud properties (Ackerman et al., 2004; Eastman and Wood, 2018; Xu and Xue, 2015). In particular, the fact that LWP increases in WEAKINV compared to CONTROL, while simultaneously decreasing in DRY_WEAKINV compared to CONTROL. indicates that not only the magnitude, but the *sign* of microphysical impacts of increased entrainment is dependent on the above-cloud environmental humidity, in line with the results of Xu and Xue (2015). In WEAKINV, the inversion strength above the

465 cloud top is reduced while the vapor mixing ratio is unchanged. In effect, this increases the above-cloud relative humidity compared to CONTROL. Despite an overall greater cloud-top entrainment rate in WEAKINV compared to control, the entrainment of higher-humidity air leads to a deepening of the cloud and boundary layer, a larger LWP, and greater cloud-top longwave radiational cooling (Figs. 2a, 4a). The deeper cloud also promotes longer-lived and larger ice particles, leading to an increase in IWP compared to CONTROL (Fig. 2b). Interestingly, while cloud depth is greater and LWP is increased in WEAKINV

compared to CONTROL, below-cloud relative humidity is reduced by a similar amount as in DRY (Fig. 10a).

In DRY_WEAKINV, by contrast, the combination of decreased inversion strength and drier above-cloud air strongly depletes cloud water compared to either CONTROL, DRY, or WEAKINV. While above-cloud relative humidity in DRY_WEAKINV is greater than in DRY, it is still reduced compared to CONTROL, while the reduced inversion strength means that this dry above-cloud air is entrained at a much greater rate than either CONTROL or DRY. Unlike WEAKINV, this increased entrainment of

475 dry air results in a dramatic reduction of cloud depth, LWP, and cloud-top longwave radiative cooling rate (Figs. 2a, 4). The entrainment of dry above-cloud air in DRY_WEAKINV also reduces the relative humidity with respect to ice in the sub-cloud boundary layer while reducing both the lifetime and depositional growth rate of ice crystals within the cloud (Fig. 10a). These effects are similar to, but more pronounced than those seen in DRY. As such, it is unsurprising that a larger proportion of sedimenting ice crystals are desiccated in DRY_WEAKINV than in either CONTROL, DRY, or WEAKINV. This increased

desiccation results in more efficient IN recycling and a greater in-cloud ice crystal number concentration in DRY_WEAKINV





compared to CONTROL, DRY, or WEAKINV (Fig. 3). As a result, by the end of the simulation, IWP in DRY_WEAKINV is similar to that in DRY and greater than that in CONTROL or WEAKINV, both of which have much less efficient below-cloud IN recycling (Fig. 2b).

The imposition of perfectly efficient IN recycling in the INFLUX simulation results in a dramatically increased IWP compared to CONTROL as a much greater number of ice crystals are present in the cloud (Fig. 2b). However, this greater ice crystal number concentration also suppresses LWP, as more supersaturation is consumed by growing ice crystals rather than by cloud droplets (Figs. 2, 12). This suppression of LWP also lessens the degree of mixing in the boundary layer; a lower LWP leads to a reduction in cloud-top radiative cooling and generation of turbulent kinetic energy, a finding supported by previous modeling studies of Arctic stratocumulus (Klein and Coauthors, 2009; Ovchinnikov et al., 2014; Solomon et al., 2018; Stevens et al., 2018) (Fig. 4). INFLUX also shows no increase in LWP or thickening of the cloud over time, unlike CONTROL, indicating that the higher ice crystal number concentration in this simulation compared to CONTROL interrupts the LWP-radiative feedback which acts to increase LWP and deepen the cloud (Figs. 2a, 3a).

Our results indicate that the recycling of IN is of key importance to the microphysical properties of the cloud and is influenced both by the rate of growth of ice crystals within the cloud as well as the characteristics of the sub-cloud boundary layer. While numerous modeling studies have previously highlighted the importance of IN recycling to the maintenance of Arctic mixed-phase clouds, the influence of above-cloud relative humidity on IN recycling has not been specifically examined (Ahola et al., 2020; Fan et al., 2009; Raatikainen et al., 2022; Solomon et al., 2015).

While surface emission rates of IN in the wintertime Arctic over sea ice appear to be quite low (Bigg, 1996; Creamean et al., 2018; Tan et al., 2023; Wex et al., 2019), the lack of any surface source of IN in our simulations may not be entirely reflective of the aerosol environment off the coast of Alaska in late April; more recent research has highlighted the role of small leads of open water as source of marine IN (Creamean et al., 2022; Griesche et al., 2021). Previous modeling of the ISDAC Flight 31 cloud also highlighted the importance of marine INP emissions to maintaining mixed-phase cloud properties over a 16 hour simulation (Raatikainen et al., 2022), as IN recycling was insufficient to preserve a steady-state IN or ice crystal number concentration, similar to our results. Further constraints on the rate of terrestrial and marine INP emissions in Arctic spring are necessary to accurately capture the influence of these IN sources in modeling of Arctic mixed-phase stratocumulus.

This study also did not account for multiple types of IN; while nucleation rates in the LCM are probabilistic, the parameterization used is that of Wex et al. (2015) for fragments of SNOWMAX bacteria, which nucleate at high temperatures. Under long-range transport of other aerosols such as mineral dust or volcanic ash, however, the omission of these INP sources may underestimate ice crystal number concentrations within the cloud (Prenni et al., 2009). Additionally, Fu and Xue (2017) found that cloud-top radiative cooling can act to nucleate colder-temperature INPs over time, helping counteract the effect of IN depletion through precipitation scavenging. Future simulations with multiple IN species would help illustrate the importance of this effect under different background aerosol and environmental conditions.

As we noted in the Introduction, several ice growth mechanisms which affect mixed-phase Arctic stratocumulus are not represented in these simulations, particularly riming and ice-ice collisions, meaning that the results of our simulations may not be exactly representative of real-world conditions. Our simulations also only examined two very specific environments,



while Arctic stratocumulus clouds are known to occur under a wide range of atmospheric conditions (de Boer et al., 2009; McFarquhar et al., 2011; Griesche et al., 2021; Shupe et al., 2006, 2011; Verlinde et al., 2007). How IN recycling affects Arctic stratocumulus clouds under such environmental conditions is therefore left unanswered by the present analysis, but may have significant effects on the structure and longevity of these clouds, as well as their radiative impacts at the surface. To address these limitations of our analysis, future modeling research with different microphysical schemes and with different initial environmental conditions, as well as more observational studies measuring IN and ice crystal number concentration within and below Arctic stratocumulus, are therefore essential to improve knowledge of the processes affecting and behavior of Arctic mixed-phase stratocumulus.

## Appendix A: Supplemental Figures

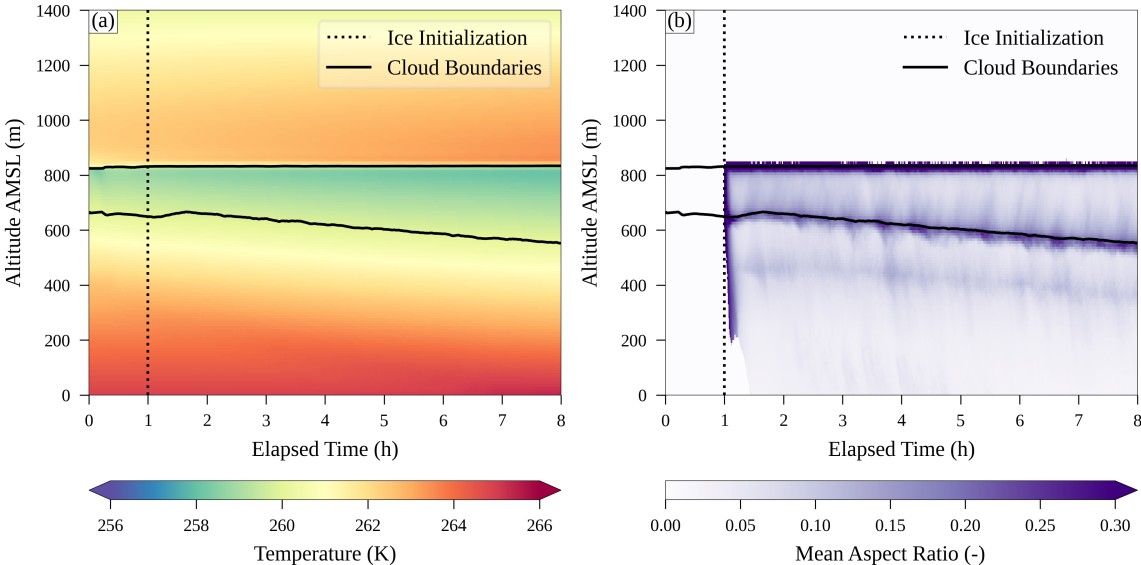

**Figure A1.** Time-height diagram of air temperature (a) and number concentration-weighted mean particle aspect ratio (b). Cloud boundaries are denoted by black lines.





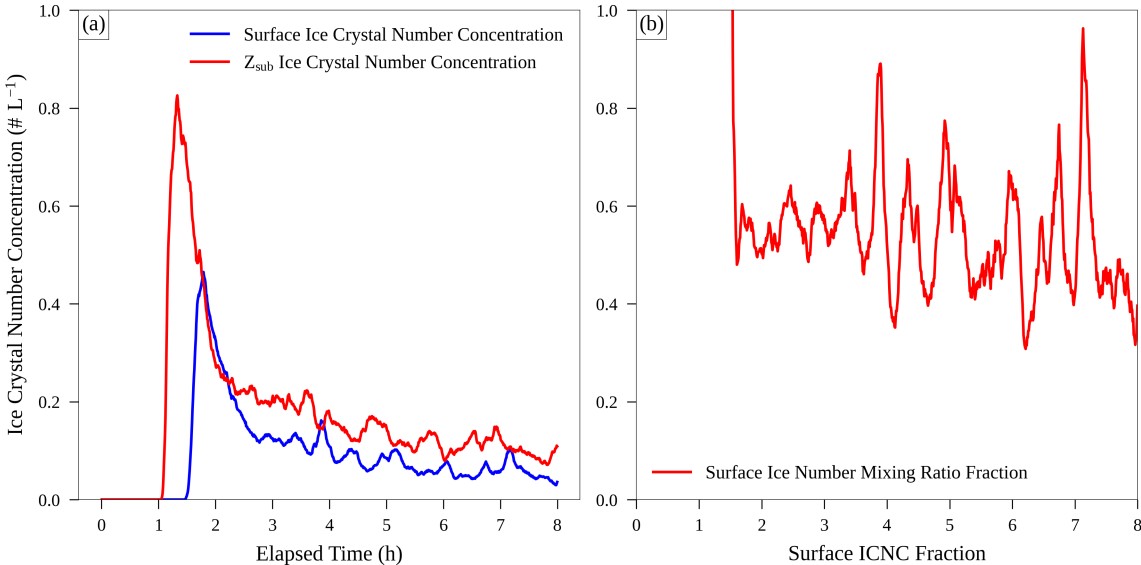

**Figure A2.** Time series of ice crystal number concentration (a) and fraction of ice crystals at the lowest ice-supersaturated altitude which reach the surface (b) in CONTROL. In (a), ice crystal number concentration at the lowest ice-supersaturated altitude is shown in red, while the ice crystal number concentration at the surface is shown in blue. In (b), the fraction of ice crystals reaching the surface is calculated using time-shifted arrays, with the shift equal to the maximum temporal correlation between the arrays of ice crystal number concentration at the lowest ice-supersaturated altitude and ice crystal number concentration at the surface.




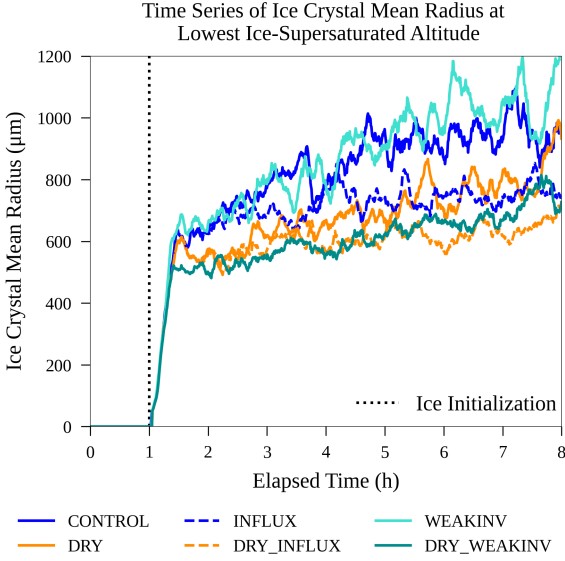

**Figure A3.** Time series of mean ice crystal radius (calculated along the maximum dimension of each ice crystal) at the lowest ice-supersaturated altitude in all six simulations.

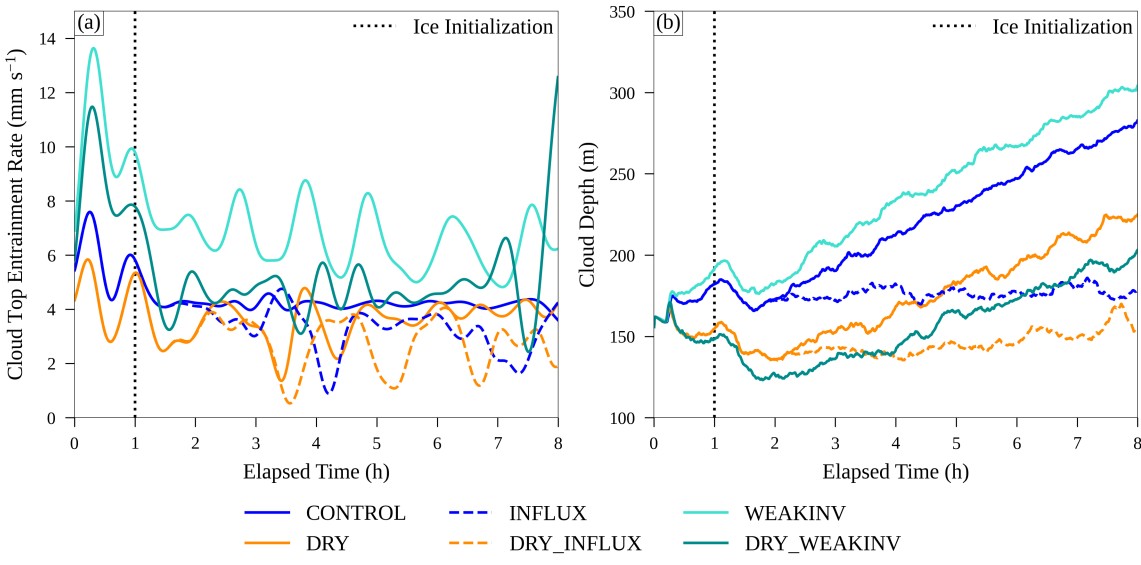

**Figure A4.** Time-series of cloud-top entrainment rate (a) and cloud depth (b) for all six simulations. Cloud-top entrainment rate is calculated the rate of change of cloud top height minus the value of large-scale subsidence.



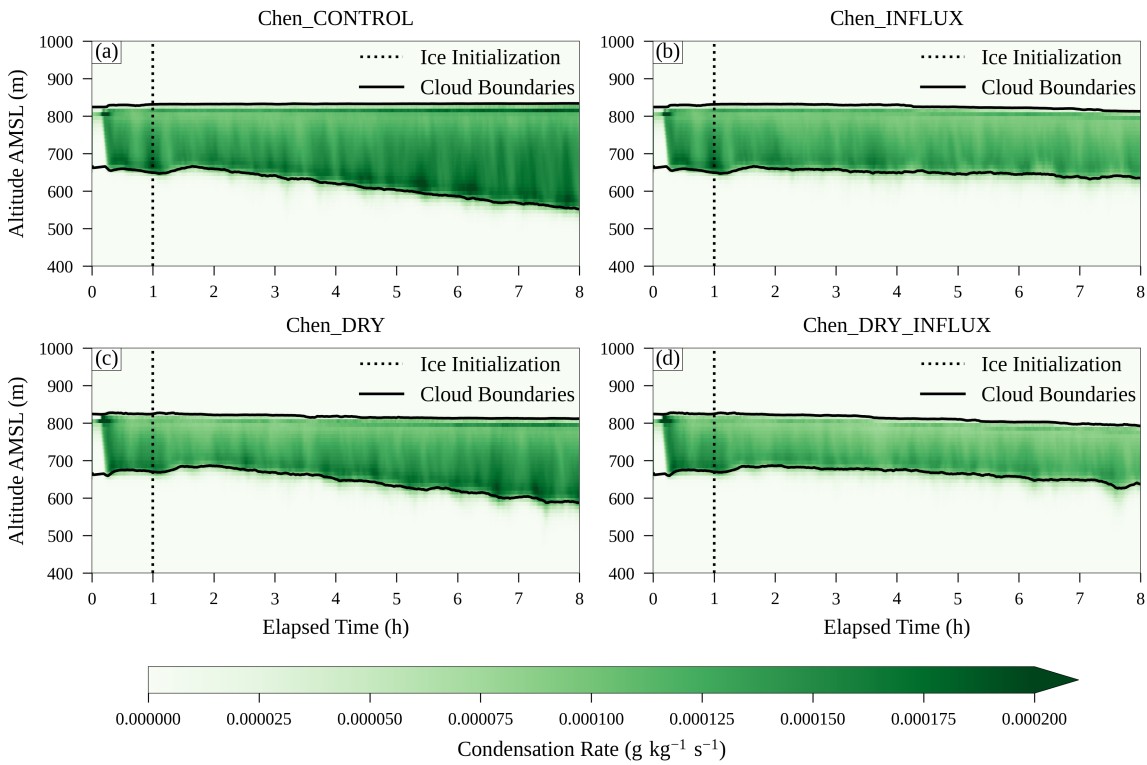

**Figure A5.** Time-height plots of total condensation rate for the CONTROL(a), INFLUX(b), DRY(c), and DRY_INFLUX(d) simulations.

*Code and data availability.* The model data used in the production of this paper is shared at the following DOI: https://doi.org/10.5282/ubm/data.720.
A Jupyter Notebook file to reproduce the figures and quantitative analysis seen in this paper is available at the following DOI: https://doi.org/10.5281/zenode
(Ascher and Hoffmann, 2025; Ascher, 2025). Ascher and Hoffmann (2025) also includes a short guide on how to set up a Python environment
and use the Jupyter notebooks of Ascher (2025).

*Author contributions.* Benjamin Ascher contributed research and the main writing component of this manuscript. Fabian Hoffmann was the
530 primary architect of the Langrangian cloud microphysics model used in this research project, offered helpful edits and contributions to the
text, and served as PhD advisor to the first author.

*Competing interests.* The authors declare that they have no conflicts of interest.



*Acknowledgements.* This work was supported by the Emmy Noether program of the German Research Foundation (DFG) under Grant HO 6588/1-1. The authors gratefully acknowledge the Gauss Centre for Supercomputing e.V. (http://www.gauss-centre.eu) for helping this project by providing computing time on the GCS supercomputer SuperMUC-NG at the Leibniz Supercomputing Centre (http://www.lrz.de).



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
