# Peer review of "Impacts of Environmental Conditions and Ice Nuclei Recycling on Arctic Mixed-Phase Cloud Properties"

_EGUsphere, 2025_

## Referee Comment (RC1)

Review of Impacts of Environmental Conditions and Ice Nuclei Recycling on

Arctic Mixed-Phase Cloud Properties, by Asher and Hoffmann, EGUSPHERE 2025-5974

In. this study, different sensitivity experiments are run for Arctic stratocumulus clouds at a temperature of -13-15C. Simulations are run using a single-layer mixed-phase Arctic stratocumulus cloud field with the System for Atmospheric Modeling, applied to the ISDAC field campaign.

Going back to the McFarquhar (2011) and other ISDAC studies, I extracted the following. Measurements of both size-resolved and bulk cloud parameters were made from 20 instruments. Aerosol size, composition, concentration, morphology, and optical and nucleating properties were measured. A continuous flow diffusion chamber (CFDC) measured IN concentrations.

Here's the assumption of your IN composition and activity. The probability of a given Super Droplet freezing is calculated using a parameterization based on the behavior of SNOWMAX bacteria, which become active INP at approximately -5C and maintain approximately the same ice nucleating activity between -10 and -20C.

The IN activity/composition measurements from ISDAC should have been used in your simulations. I'm concerned about how the IN were activated in cloud. They are all activated at the same time, one hour after cloud initialization. This is very unrealistic as there is a range of activation of the IN, even at a given temperature. Some of the ISDAC in-situ measurements could have been used for this purpose.

I would have liked to see plots of the air vertical velocity with your different cases.

These are the main comments I have now, more to come.